# Co-Functionalization of Gold Nanoparticles with C7H2 and HuAL1 Peptides: Enhanced Antimicrobial and Antitumoral Activities

**DOI:** 10.3390/pharmaceutics14071324

**Published:** 2022-06-23

**Authors:** Daniela M. D. Formaggio, Jéssica A. Magalhães, Vitor M. Andrade, Katia Conceição, Juliana M. Anastácio, Gabrielli S. Santiago, Denise C. Arruda, Dayane B. Tada

**Affiliations:** 1Laboratory of Nanomaterials and Nanotoxicology, Institute of Science and Technology, Federal University of São Paulo (UNIFESP), São José dos Campos 12231-280, SP, Brazil; dformaggio@gmail.com (D.M.D.F.); jessica.magalhaes@unifesp.br (J.A.M.); gabriellissantiago@outlook.com (G.S.S.); 2Laboratory of Peptide Biochemistry, Institute of Science and Technology, Federal University of São Paulo (UNIFESP), São José dos Campos 12231-280, SP, Brazil; vitormartinsdeandrade@yahoo.com.br (V.M.A.); katia.conceicao@unifesp.br (K.C.); 3Laboratory of Experimental Cancer Biology, University of Mogi das Cruzes (UMC), Mogi das Cruzes 08780-911, SP, Brazil; juliiana_machado@hotmail.com

**Keywords:** gold nanoparticles, therapeutic peptides, HuaL1, C7H2, CDR peptides, tumor targeting, metastatic melanoma

## Abstract

The functionalization of nanoparticles with therapeutic peptides has been pointed out as a promising strategy to improve the applications of these molecules in the field of health sciences. Peptides are highly bioactive but face several limitations such as low bioavailability due to the difficulty of overcoming the physiological barriers in the body and their degradation by enzymes. In this work, gold nanoparticles (AuNPs) were co-functionalized with two therapeutic peptides simultaneously. The peptides from the complementary determining region of monoclonal antibodies, composed of the amino acid sequences YISCYNGATSYNQKFK (C7H2) and RASQSVSSYLA (HuAL1) were chosen for having exhibited antitumor and antimicrobial activity before. The peptides-conjugated AuNPs were characterized regarding size, morphology, and metal concentration by using TEM, dynamic light scattering, and ICP-OES techniques. Then, peptides-conjugated AuNPs were evaluated regarding the antimicrobial activity against *E. coli*, *P. aeruginosa*, and *C. albicans*. The antitumoral activity was evaluated in vitro by cell viability assays with metastatic melanoma cell line (B16F10-Nex2) and the cytotoxicity was evaluated against human foreskin fibroblast (Hs68) cell line. Finally, in vivo assays were performed by using a syngeneic animal model of metastatic melanoma. Our findings have highlighted the potential application of the dual-peptide AuNPs in order to enhance the antitumor and antimicrobial activity of peptides.

## 1. Introduction

Metallic nanoparticles have been extensively explored in biomedical research as drug delivery systems, imaging agents, and therapeutic agents, as, for example, in photothermia [1,2,3,4,5]. The main advantages of metallic nanoparticles rely on their optical properties, low toxicity and immunogenicity, good biocompatibility, and excellent stability. These nanoparticles are also easily conjugated with biological and non-biological molecules providing the most diverse functionalities [1,6,7,8,9]. Amongst the metallic nanoparticles, gold nanoparticles (AuNPs) stand out as the most investigated in the last decades. AuNPs have been pointed out as potential agents for imaging, sensoring, photothermia, and antimicrobial applications providing suitable therapeutic efficiency and low toxicity [10,11,12,13]. The AuNPs have also been reported as useful antitumoral therapeutics, for example, against breast cancer cells. These NPs have shown an ability to inhibit angiogenesis, a process that plays a crucial role in cancer progression [14]. These characteristics make them excellent delivery vehicles to tumoral cells promoting more efficient treatment [15]. Currently, there is not a 100% effective treatment for curing cancer, and nanotechnology has been seen as a potential tool for the treatment of this disease [3,16,17].

Additional to nanotechnology, the search for innovative therapeutics for cancer treatment has also highlighted the potential application of therapeutic peptides. According to recent reports, there are more than 100 peptide-based drugs that are found commercially available, and another 688 in clinical screening [18,19]. The main challenging aspect in the translation of peptides in clinical application for cancer treatment is that most peptides, especially naturally occurring ones, exhibit low bioavailability as they face several physiological barriers in the body. In addition, they are likely to be deactivated by plasma proteins or proteolytic enzymes [19,20,21,22].

Previously, Arruda, D.C., and collaborators have investigated different peptide sequences extracted from determining regions of the Complementarity-determining regions (CDRs) of monoclonal antibodies [20,21,22]. Several peptides were shown to display outstanding antitumoral, antimetastatic, and antimicrobial activity. The peptides termed C7H2 and HuAL1, with their respective amino acid sequences, YISCYNGATSYNQKFK and RASQSVSSYLA, significantly reduced the growth of metastatic melanoma cells (B16F10-Nex2), triggering caspase-dependent apoptosis. Both sequences also presented anti-infective activities against different types of microorganisms, including viruses and fungi [23,24]. Nevertheless, the application of these peptides in a clinical application would also be hampered by the low stability of these peptides in the biological medium, which would require the peptides to be used at high concentrated doses. This limitation would increase the costs considering the high cost of peptides synthesis and could also compromise the therapeutic efficiency since it could increase collateral effects.

In this context, the present work addressed the strategy of using AuNPs as a drug delivery platform in order to provide enhanced tumor targeting and to protect the peptides from degradation by biomolecules. The use of nanoparticles as delivery agents for peptides has already been reported by different researchers and also by our group [25,26,27,28]. Recently, the encapsulation of peptides into polymeric nanoparticles has been shown to be an efficient strategy to enhance the antimetastatic activity of therapeutic peptides [29]. Herein, the use of AuNPs was further explored since it was used not only to avoid peptide degradation but also to combine two peptides, providing a synergistic activity of both HuaL1 and C7H2 peptides. The performance of AuNPs as a dual-peptide platform was evaluated regarding the antimicrobial activity against *E. coli*, *P. aeruginosa,* and *C. albicans* antitumoral activity in vitro by cell viability assays with metastatic melanoma cell line (B16F10-Nex2) and with human foreskin fibroblast (Hs68) cell line and by in vivo assays in an animal model of metastatic melanoma. Our findings confirmed that AuNPs have great potential as antitumor peptide delivery systems since they can overcome the pharmacokinetic limitations inherent to these molecules.

## 2. Materials and Methods

### 2.1. Materials

Tetrachloroauric acid (HAuCl4.3H2O; ≥99.9% trace metals basis), sodium citrate monobasic (Na3C6H5O7; purum p.a., anhydrous, ≥99.0%), mercaptoundecanoic acid (MUA; 95%), N-(3-dimethylaminopropyl)-N′-ethylcarbodiimide hydrochloride (EDC; purum, ≥98.0%) and N—hydroxisuccinimide (NHS; 98%) were purchased from Sigma-Aldrich. 2-(N-morpholino) ethanesulfonic acid (MES; 99%) was supplied by Vetec. All the samples were prepared by using deionized water (Direct Q 3 UV-MilliQ, Merck Millipore; Darmstadt, Germany). The peptides C7H2 (YISCYNGATSYNQKFK) and HuAL1 (RASQSVSSYLA) were supplied by Peptide 2.0 (≥95% of purity). RPMI (Gibco) and DMEM (Gibco) medium were prepared in deionized water. Fetal bovine serum (FBS), streptomycin, and ampicillin were used to supplement the medium and were supplied by Cultilab and ThermoFisher Scientific, respectively. For cell viability assay, it was used trypan blue (solution 0.4%) and trypsin, both purchased from Sigma Aldrich. The antimicrobial studies were carried out in agar Muller-Hinton (Kasvi) and agar Sabouraud (Kasvi). Brain Heart Infusion (BHI, KASVI) was used to prepare pre-inocula of the strains. The in vivo experiments were carried out according to the Animal Use Ethics Committee (CEUA) of the Federal University of São Paulo (ethical approval code: 2210300817; 8-11-2017).

### 2.2. Synthesis of AuNPs

AuNPs were synthesized according to the Turkevich method with adaptations as already reported in our previous work [10]. Firstly, 20 mL of an aqueous solution of HAuCl_4_.3H_2_O at 50 mM was added to 1 L of ultrapure water (type 1) at 95 °C. Then, 10 mL of an aqueous solution of sodium citrate (Na_3_C_6_H_5_O_7_) at 0.3 M was added. Na_3_C_6_H_5_O_7_ was used as an Au^3+^ reducing agent and as the NP’s stabilizer agent. The NP’s suspension was kept under magnetic stirring until it turned intensely red-colored. The final AuNPs suspension was cooled at room temperature and stored at 4 °C.

### 2.3. AuNPs Functionalization with Peptides

The peptides-conjugated AuNPs were prepared by using an adaptation of the protocol reported by Kwon and co-workers [30]. At first, 1.4 mL of an ethanolic solution of MUA (10 mM) were added into a 10 mL of AuNPs suspension. The mixture was kept under mechanical stirring (hand-made equipment developed by Forgers group (ICT-UNIFESP) for about 40 min. The suspension was stored at room temperature for 3 h. After that, AuNPs were centrifuged (21,380× *g*, 20 min, 15 °C) and the pellet was resuspended in MES buffer at 10 mM (pH = 6.0) by using an ultrasound bath. Subsequently, 1.4 mL of EDC (10 mM) and NHS (10 mM) solutions prepared in MES buffer were added into AuNPs suspension. The suspension was stirred for about 15 min in order to provide the NP’s surface reaction with the crosslinkers. As described by Xie et al. [31], the use of EDC/NHS results in the amine-reactive ester that can react with the primary amino group of peptides. After this period, the peptides were added at the concentrations described in Table 1 and the suspension was stirred for 12 h. Finally, peptides-conjugated AuNPs (Figure 1) were centrifuged twice (21,380× *g*, 20 min, 15 °C) to wash out the excess free peptides in solution. At this stage, there was an attempt to quantify the peptides in the supernatant by using a BCA kit (Sigma). However, no signal was observed and therefore the concentration was probably too low to be detected by this colorimetric method. Therefore, this result was considered indicative that almost all the peptides used in the functionalization had been conjugated to the AuNPs. AuNPs conjugated to HuAL1 and C7H2 peptides individually (**H-AuNPs and C-AuNPs**) and conjugated to both peptides (**HC-AuNPs**) were stored at 4 °C for further analysis.

### 2.4. UV-Vis Spectra of AuNPs before and after Conjugation with Peptides

The conjugation of AuNPs with HuAL1 and C7H2 peptides was monitored by UV-Vis spectroscopy (Spectrophotometer V-730, Jasco, Easton, MD, USA). Deionized water was used as a reference sample. The measurements were performed in the range of 200–800 nm with a step of 10 nm and a rate of 400 nm/cm. The samples were evaluated without any dilutions after the functionalization steps.

### 2.5. Dynamic Light Scattering (DLS)

The changes in size and zeta-potential of AuNPs after the functionalization with the peptides were evaluated by using the DLS technique in a DelsaNano C, Beckman Coulter equipped with a 630 nm laser light, available at NAPCEM ICT-UNIFESP. The samples were diluted in type I deionized water. Values are depicted as mean values and standard deviation (*n* = 3). The autocorrelation function CONTIN and the Stokes-Einstein equation (Equation (1)) were used to obtain the particle size distribution and the values of hydrodynamic diameter, respectively.
(1)$$Z=\frac{ƞ}{\varepsilon_{0}\varepsilon_{r}}U$$
where ƞ is the refractive index of a medium, $\varepsilon_{0}$ is the dielectric constants in the vacuum, $\varepsilon_{r\;}$ is the dielectric constants in the vacuum of the solvent and U is the electrophoretic mobility. The values of zeta-potential were calculated from the Smoluchowski equation (Equation (2)). The Smoluchowski equation was chosen because although the particles were smaller than 100 nm, the ionic strength of the medium was higher than 1 mM.
(2)$$\nu_{D}=\frac{U\cdot ƞ}{\lambda }sin\;\theta$$
where *λ* is the wavelength of incident light and *θ* is the scattering angle.

### 2.6. Transmission Electron Microscopy (TEM)

The size and morphology of AuNPs were analyzed by TEM. The NPs’ suspension was dropped on a carbon-coated copper grid and dried at room temperature. A microscope FEI TECNAI G2 TF20 XT available at the Federal University of São Carlos (UFSCAR/FAPESP 2013/07296-2) was used to characterize the size and morphology of the NPs at 120 kV. The TEM images were analyzed by ImageJ software to calculate the mean diameter and size distribution (*n* = 45).

### 2.7. Inductively Coupled Plasma Optical Emission Spectroscopy (ICP-OES)

The metal concentration and composition of the NPs were evaluated by ICP-OES. The samples were previously centrifuged and resuspended in deionized water, without dilution or drying. The analysis was performed by a spectrometer Arcos with radial-view (Spectro, Inc., Chelmsford, MA, USA) at Central Analítica (Institute of Chemistry, University of São Paulo).

### 2.8. Cell Viability Assays

The cytotoxicity of peptides free in solution or conjugated to AuNPs was evaluated by the trypan blue exclusion test. This assay is based on the principle that live cells possess an intact membrane, which is impermeable to the trypan blue dye. On the other hand, dead cells have damaged membranes, allowing the internalization of the dye into the cell, making its cytoplasm blue. In this way, the cytotoxicity of peptides-conjugated NPs was monitored by using non-tumoral cell line Hs68 and metastatic melanoma cell line B16F10-Nex2 (*n* 0342-BCRJ, Rio de Janeiro, Brazil). Human skin fibroblasts Hs68 cells (ATCC CRL-1635™) were cultured in DMEM medium supplemented with 10% (*v*/*v*) FBS and the antibiotics streptomycin (0.1 g/L) and ampicillin (0.025 g/L). Cells were incubated in a humidified atmosphere with 5% CO_2_ at 37 °C. Hs68 cells were seeded into 96-well plates (1 × 10^4^ cells per-well) and incubated for 24 h. B16F10-Nex2 cells were cultured in RPMI medium, also supplemented with 10% (*v*/*v*) FBS and the antibiotics streptomycin (0.1 g/L) and ampicillin (0.025 g/L). B16F10-Nex2 cells were seeded into 96-well plates (5 × 10^3^ cells per-well) and incubated for 24 h in a humidified atmosphere with 5% CO_2_ at 37 °C. Afterward, the respective culture medium was removed from each cell plate and replaced by a fresh culture medium containing different samples (Table 2), as well as HuAL1 and C7H2 (0.6 mM) free in solution. After 24 h, the culture medium was removed, and each well was washed twice with fresh PBS. Cells were collected by using trypsin and counted in a Neubauer chamber, using trypan blue dye and an optical microscope (TCM 400, Labomed, Los Angeles, CA, USA). Cells incubated in the absence of samples were considered as 100% of viability. The experiments were performed in triplicate. The results were expressed as the average value of % of cell viability and standard deviation.

### 2.9. Antimicrobial Studies

The antimicrobial activity of peptides-conjugated NPs was monitored by liquid growth inhibition assay against *Staphylococcus aureus* (ATCC 6538), *Pseudomonas aeruginosa* (ATCC 15442), and *Candida albicans* (ATCC 10231). The protocol used herein was an adaptation of the protocol reported in our previous work [10]. Pre-inocula of the strains were prepared by using a BHI medium. Peptides-conjugated NPs (Table 3) were added to 96-well plates with 100 µL of a microorganism suspension (10^6^ CFU/mL) in each well. BHI medium was also used to complete the volume of each well until it reached 200 µL. The microorganisms were incubated at 37 °C for 24 h. A suspension of microorganisms in the absence of NPs was used as a positive growth control. After the incubation period, the microbial growth was measured by optical density (OD) by using a Synergy plate reader (BioTek Instruments, Santa Clara, CA, USA) at 630 nm. The values of OD were compared to the inoculum growth (positive control). The information described in Table 3 was used to evaluate each one of the peptides-conjugated NPs (H-AuNPs, C-AuNPs, HC-AuNPs).

### 2.10. Antitumoral In Vivo Assay

C57BL/6 7-8-weeks-old mice were intravenously injected with 5 × 10^5^ syngeneic B16F10-Nex2 melanoma cells per mouse (0.1 mL). Subsequently, the animals were intraperitoneally treated with 0.1 mL of HC-AuNPs (50 µg of each peptide) or with 0.1 mL of H-AuNPs (50 µg of HuaL1). As a positive control, the animals were treated with 0.1 mL of HuAL1 in PBS (50 µg) or with 0.1 mL of both HuAL1 and C7H2 (50 µg of each peptide). Mice were also treated with 0.1 mL of unconjugated AuNPs in PBS as a negative control. The samples were injected on alternate days after tumor induction, until completing six days of treatment. After 21 days, the animals were euthanized, and their lungs were collected and inspected for metastatic colonization. The experiment was performed in quintuplicate. The animal experiments were approved by the Ethics Committee on the Use of Animals of UNIFESP and UMC (ethical approval code: 2210300817 8-11-2017 and 020/2016 17-08-2016). The results were expressed as the % of the tumoral area measured by the analysis of lung images in the ImageJ software.

### 2.11. Statistical Analysis

One way ANOVA (*p* < 0.001) was used to compare data. Considering that the differences in the means were significant, a post hoc pair-wise comparison was conducted using Tukey-Kramer’s test. Values were depicted as mean values and standard deviation.

## 3. Results and Discussion

### 3.1. Physicochemical Properties of AuNPs before and after Functionalization with Peptides

Considering the suitable performance for biological application of AuNPs observed in our previous work, herein AuNPs were chosen as a platform for peptides delivery aiming antitumoral and antimicrobial therapies. AuNPs were characterized regarding their size and morphology by TEM (Figure 2). The image showed spherical and homogeneous size distribution of NPs, with a mean diameter of 21 ± 5 nm. The AuNP’s suspension was further characterized regarding the total amount of gold by ICP-OES, which indicated that the suspension contained 1.9 mM of gold.

Since AuNPs were supposed to be used in colloidal suspension, the hydrodynamic diameter was also measured (Table 4). The DLS measurements indicated that the AuNPs had 23 ± 8 nm of hydrodynamic diameter, which was in agreement with the NPs’ size measured by TEM. After peptides conjugation, there was an increase of more than 10 times in the hydrodynamic diameter of the NPs. These modifications on the surface of the NPs were indicative that the coupling of peptides to the NPs was successful. In the same way, it was observed that the zeta-potential values changed from very negative to low positive values after AuNPs functionalization with peptides. Although it could suggest a loss of colloidal stability, this could be an advantage, since positive NPs can modulate cell-membrane potential, leading to an enhancement of cellular uptake and of cytotoxicity [32].

The peptides conjugation to AuNPs was also monitored by UV-vis spectroscopy (Figure 3). The UV-vis spectrum of AuNPs before conjugation showed a characteristic plasmon band (λ_max_) at 520 nm. According to Haiss [33], this value is associated with NPs of about 21 nm, which is also in agreement with the values obtained by TEM and DLS. In addition, according to the same reference, the molar decadic extinction coefficient (ε450) was 2.67 × 10^8^ mol of NPs/L × cm^−1^. Therefore, in the used suspension of AuNPs, the concentration was 0.46 × 10^−8^ mol/L of NPs. After coupling the peptides to the NPs, it was noticed a broadening of the λ_max_, which may indicate the presence of the peptides HuAL1 and C7H2. Moreover, the peptides conjugation was also evidenced by the decrease in intensity and the shift of the λ_max_ to longer wavelengths. These observations may be a result of the changes in the refractive index of the medium around the NPs, and they may occur due to an electronic interaction between the peptides and the NPs’ surface [34]. It has to be pointed out that the broadening and red-shift of the plasmon absorbance band could also be a result of NPs aggregation.

### 3.2. Cytotoxicity of Free Peptides and Peptides-Conjugated AuNPs

The antitumoral activity of therapeutic peptides depends on several factors, including their concentration, solubility in the biological medium as well as their biodistribution. A better performance would be expected to involve high cytotoxicity against tumor cells and low cytotoxicity against non-tumoral cells. Therefore, the peptides-conjugated NPs studied herein were evaluated against murine metastatic melanoma (B16F10-Nex2) and non-tumoral human foreskin fibroblast (Hs68) cell lines. For comparative purposes, the cytotoxicity of free peptides in solution was also evaluated. As expected, AuNPs did not show cytotoxicity to Hs68 cells, indicating the high biocompatibility (77%) of these NPs (Figure 4). Nevertheless, both free peptides in solution and conjugated to NPs were cytotoxic.

Despite the slight increase of viability of the cells incubated with peptides-conjugated NPs, the viability of cells was not statistically different from the viability of cells incubated with the peptides in solution. The viability of Hs68 cells incubated with H-AuNPs (39%), and C-AuNPs (54%) was lower than 70%, indicating their cytotoxicity against Hs68 cells. Some factors that may contribute to this observed cytotoxic effect are the concentration and the incubation time of both peptides. According to Polonelli and co-workers [24], the concentration of HuAL1 and C7H2 peptides that inhibits by 50% the viability of B16F10-Nex2 cells (IC50) was 0.6 mM and 0.05 mM, respectively. In this work, the concentration of peptides that was chosen for the in vitro experiments (0.6 mM) was about twelve times higher than the IC50 for the C7H2 peptide. This way, the toxic effect of C7H2 and HuAL1 peptides on Hs68 cells may be attributed to their high concentration. In addition, in Arruda et al.’s previous work the cytotoxicity of C7H2 to non-tumor cell lines was evaluated [20]. Their results revealed that not only the concentration of peptide but also an extended incubation time may significantly inhibit cellular growth.

When both peptides were simultaneously used the cytotoxic effect was increased. Although HC-AuNPs have shown a cytotoxic effect, these NPs showed a lower cytotoxic effect against Hs68 cells than the combination of both peptides free in solution. The viability of cells incubated with HC-AuNPs (40%) was 2 times higher than the viability of cells incubated with both peptides free in solution (20%). Clearly, when used in solution, the peptides may have compromised the cellular growth due to the synergistic effect of using both peptides together. Moreover, it is noteworthy that the use of AuNPs to conjugate both peptides had an important role in decreasing the cytotoxicity of the peptides against non-tumoral Hs68 cells. This feature is extremely important since the effectiveness of therapeutic platforms depends on minimum toxicity to healthy cells and a more pronounced toxicity to tumor cells.

The antitumoral activity of both peptides in solution and conjugated to AuNPs was evaluated against B16F10-Nex2 cells (Figure 5). Similar to what was observed with Hs68 cells, AuNPs did not show cytotoxicity against B16F10-Nex2 cells (89% of cell viability).

The viability of B16F10-Nex2 cells after incubation with free peptide C7H2 in solution was close to the value found after incubation with C-AuNPs. In both cases, a satisfactory cytotoxic effect on the peptide, reducing the viability of the tumor cells to 13% and 18%, respectively, was noted. On the other hand, the viability of cells incubated with H-AuNPs was 3 times lower than the viability of cells incubated with the peptide-free HuAL1 in solution. This enhancement of cytotoxicity by using NPs was consistent with other studies reported before, which demonstrate that the use of NPs can increase the effectiveness of drugs and/or molecules when compared to their free form [35,36]. Comparing both types of peptide-conjugated AuNPs, the cytotoxic effect of H-AuNPs was 1.4 times higher than the one of C-AuNPs. This observation can be assigned to the tested concentration of the peptides. The concentration of peptides used herein in this work was close to the reported IC50 of HuaL1, whereas the concentration of C7H2 was much higher than its IC50. That is the reason why the cytotoxic effect of C7H2 was maximal at the tested concentration, so that the effect of its conjugation to NPs could not be evidenced.

The cytotoxicity effect of peptides-conjugated NPs was even more evident after the simultaneous conjugation of both peptides on the AuNPs (HC-AuNPs). The viability of cells incubated with HC-AuNPs was 21% whereas the viability of cells incubated with free HuAL1 and C7H2 peptides in solution was 69%. Therefore, HC-AuNPs showed a cytotoxicity 3 times higher than the mixture of both peptides in solution. It is noteworthy that, in this case, the tested concentration of each peptide that was conjugated to the NPs was 0.3 mM. This way, the results demonstrated that the functionalization of AuNPs with both peptides promoted an increase in their antitumoral activity and lower cell viability could be observed even by using a concentration 2 times lower than the concentration of the individual peptides previously tested. This observation is in line with recent studies, that showed an interesting improvement in the antitumoral activity of peptides due to their conjugation to NPs [25,27,28,29]. Notoriously, the conjugation of both peptides to AuNPs was proven herein to successfully decrease the cytotoxicity against non-tumoral Hs68 cells and at the same time increase the cytotoxicity against tumor B16F10-Nex 2 cells.

### 3.3. Antitumoral In Vivo Evaluation

It is noteworthy that the advantages of using NPs as drug carriers may be more evident in vivo since the conjugation of peptides and drugs to NPs can improve tumor targeting and avoid the biodegradation of the conjugated molecules. In this way, considering the encouraging in vitro results, the antitumoral effect of the peptides-conjugated NPs was further evaluated by using an animal model of metastatic melanoma. After the treatment of the animals with the free peptides in solution and peptides-conjugated AuNPs, the lungs were extracted in order to evaluate the incidence of melanoma metastasis. The image of the extracted lungs from the animals and the tumoral area calculated after the treatment is shown in Figure 6.

Enhanced antitumoral activity of both HuAL1 and C7H2 was observed after their conjugation to NPs. The tumoral area after the treatment with HC-AuNPs was 2%, whereas after the treatment with both peptides free in solution was 12%. On the other hand, there was no statistical difference between the values of the tumoral area after the use of free HuAL1 in solution and conjugated to AuNPs (H-AuNPs). The resultant tumoral area of the lungs extracted from the animals treated with the peptide HuAL1, as well as those treated with H-AuNPs, was 4%. It is important to highlight that the dose of each peptide used herein was about 5 times lower than the dose used by Polonelli and co-workers, who observed an antitumoral effect of both peptides free in solution. That is an interesting result, since the use of peptides at high concentration may increase side effects, and one of the main goals of researchers in the field of health sciences is to achieve high therapeutic efficiency at a low concentration of drugs. Therefore, it is expected that the use of peptides-conjugated AuNPs enhances the efficiency of cancer treatment by promoting high anti-tumoral activity at a low concentration of peptides. In this way, although H-AuNPs did not show any improvement in the antitumoral activity of the HuAL1, HC-AuNPs showed an enhancement of the antitumoral activity of both peptides when used simultaneously. When compared with the use of both free HuAL1 and C7H2 in solution, HC-AuNPs were capable to inhibit 81% of the lung metastasis. This could be afforded by the synergistic effect observed after the conjugation of both peptides to AuNPs. The use of peptides free in solution may show limited biodistribution, since peptides may react to each other and with biomolecules, causing its aggregation and degradation before reaching the tumor site. On addition, the delivery of the two peptides to the tumor site could not be guaranteed to be simultaneous by injecting free peptides in solution since each one would have particular pharmacokinetics. The combination of these factors may lead to the low efficiency of cancer treatment with peptides in solution despite the very intense activity observed in vitro. In this context, the HC-AuNPs presented here can bring the advantages of protecting the peptides from aggregation and degradation and improving their circulation time. In addition, our findings pointed out that the use of AuNPs to conjugate peptides may be a clever strategy to promote cellular and tumor targeting and guarantee the simultaneous delivery of the peptides to the tumor site.

### 3.4. Antimicrobial Studies

Since the peptides used herein have also been investigated as antimicrobial agents, the antimicrobial activity of peptides-conjugated AuNPs was investigated in order to evaluate the effect of conjugation on the antimicrobial property of the peptides. The antimicrobial activity of peptides-conjugated NPs was evaluated against the two strains of bacteria *S. aureus* and *P. aeruginosa*, and against the fungus *C. albicans* (Table 5; Figure 7). The concentration of each type of peptide present in the suspension of peptide-conjugated NPs used in the antimicrobial assays is depicted in Table 5.

The values of inhibition of microbial growth observed after 24 h of incubation of microorganisms with each type of peptide-conjugated AuNPs are shown in Figure 7. Differently from what was expected, it wasn’t observed a synergistic effect after a combination of the peptides HuAL1 and C7H2 in the AuNPs. HC-AuNPs showed a maximum antimicrobial activity of 34% against *S. aureus* by using an intermediary concentration of 0.6 mg/mL, whereas the concentration of 1.2 mg/mL promoted only 24% and 28% of inhibition of growth of *P. aeruginosa* and *C. albicans*, respectively. A similar antimicrobial activity was observed after incubation with C-AuNPs (0.6 mg/mL), which promoted an average of 26% of inhibition of growth of the three strains of microorganisms. On the other hand, H-AuNPs showed the highest antimicrobial activity against all the strains of microorganisms. The maximum inhibition of microbial growth was observed after using the more concentrated H-AuNPs suspension. In the incubation with *S. aureus* the H-AuNPs were used at 1 mg/mL of peptide. *P. aeruginosa* and *C. albicans* were incubated with H-AuNPs suspension containing 1.2 mg/mL of peptide. Even if the highest concentrations have been required in order to observe inhibition of microbial growth, H-AuNPs showed an antimicrobial activity approximately 2 times higher than C-AuNPs and HC-AuNPs against *S. aureus* and *C. albicans*.

According to Polonelli [24], the use of a solution at 0.1 mg/mL of the peptide C7H2 was able to inhibit 100% of the growth of the fungus *C. albicans*, whereas the same concentration of the peptide HuAL1 did not show any inhibitory activity. Herein, our results showed that some properties of the peptides, like their antimicrobial activity, may be changed after their conjugation with NPs, that is why different values of maximum inhibition of microbial growth were observed after coupling the peptides to AuNPs. This observation could be related to the mechanism of peptide binding to NPs [27]. Considering that some regions of the peptides C7H2 and HuAL1 play a key role in the mechanisms of inhibition of microbial growth, the conjugation of peptides to AuNPs by using specific sites can modulate their antimicrobial activity [24]. Herein, we suggest that the conjugation of C7H2 to AuNPs may have deactivated the site of the peptide that was responsible to compromise the microbial growth. Conversely, the conjugation of HuAL1 to AuNPs may have promoted a better antimicrobial activity by stabilizing the peptide and allowing it to expose key-amino acid residues in the H-AuNPs.

In addition to the modulation of the physicochemical and biological properties of peptides due to their interaction and binding with the surface of NPs, the metal concentration may also be an important factor in the antimicrobial response of the peptides-conjugated NPs. The correspondent metal concentration of AuNPs and the antimicrobial activity observed for peptides-conjugated NPs are shown in Table 6.

According to our previous results [10], the metal concentration of AuNPs that was able to inhibit at least 40% of microbial growth was 11.8 µg/mL against *S. aureus* and *C. albicans*, and 23.6 µg/mL against *P. aeruginosa*. However, in the present study, it was noticed that the metal concentration of peptides-conjugated NPs that was able to inhibit approximately 30% of the growth of the same microorganisms was about 10 to 20 times higher than those observed for AuNPs without any peptide. For this reason, it was also suggested that, besides the action of the peptides, AuNPs had a strong contribution to the antimicrobial activity of the H-AuNPs.

## 4. Discussion

Our results have pointed out that the functionalization of AuNPs with peptides is a clever strategy to enhance the antitumoral and antimicrobial activities and overcome the drawbacks of using therapeutic peptides in solution. Although there are many challenges related to the understanding of the pharmacokinetics of peptides-conjugated NPs, our studies have shown that the surface chemistry of the AuNPs is suitable to conjugate the peptides. Moreover, the AuNPs were able to be functionalized with two different types of peptides simultaneously, promoting a combined antitumoral activity against metastatic melanoma. Especially in vivo, the enhanced antimetastatic effect could be provided by the synergistic activity of both C7H2 and HuaL1 peptides. In addition to the well-known enhanced penetration and retention effect (EPR effect) of NPs [16,17,37], herein the antitumoral activity of the peptides is expected to be favored by the protection of peptides from degradation by the biological medium and the simultaneous delivery of both peptides to the tumor site.

The in vitro assays had shown that the functionalization of AuNPs with peptides reduced their cytotoxicity against non-tumoral cells and enhanced their cytotoxicity against tumor cells. HC-AuNPs induced 79 ± 9% of cell death in B16F10-Nex2 cells, compared to 60 ± 8% when incubated with HS68 cells. These differences in the cytotoxic effect of the peptides-conjugated NPs between the cell lines may be a consequence of the different interactions between AuNPs and each type of cell. The efficiency of absorption of free and peptides-conjugated AuNPs may vary with the composition of the cell membrane and the metabolic activity of the cells [38]. The high metabolic activity of tumor cells, as well as the altered membrane potentials, can lead to overexpression of surface receptors, in addition to disturbing proteins involved in endocytic internalization pathways, contributing to increased NP uptake [39,40,41]. These differences between tumor and non-tumor cells are the main factors that will define the route, mechanism, efficiency of absorption, and intracellular distribution of NPs.

AuNPs of different sizes and shapes (beads and rods) were already functionalized with a new pro-apoptotic peptide, with the sequence LA-WKRAKLAK, to increase their anticancer potency. The spherical and inert nature of these platforms was able to regulate the pro-apoptotic activity of these peptides, in addition to providing a greater density of these on the NP surface, increasing the bioavailability of the molecule. The authors found a 141-fold increase in cellular cytotoxicity of the LA-WKRAKLAK peptide conjugated to AuNPs in drug-resistant breast cancer cell lines (MCF-7), compared to the free LA-WKRAKLAK peptide. In another study, the antitumor peptide, cathelicidin LL-37, and its synthetic analog, ceragenin CSA-13, were covalently immobilized on the surface of magnetic NPs (MNPs) and tested against colorectal adenocarcinoma cells (DLD-1). The results revealed that both peptides alone did not significantly affect the survival of DLD-1 cells, whereas individual conjugation to NPs caused an improved and dose-dependent cytotoxic response [42]. According to Kluwe, L. [43], tumor cell culture and non-tumor ones provide a potential resource in vitro to assess the specificity or selectivity of a given compound, which may clarify its clinical toxicity. Although some of the aforementioned works describe the cell type as an important factor in the NPs’ cell internalization process, few of them compared the NP internalization in the different cell lines. Our results are in line with the previous works and they suggest that the functionalization of AuNPs with peptides can modify their cytotoxicity and that cell type is a critical factor for this effect.

The promising results observed in vitro with the peptide HuAL1 and the mixture of HuaL1/C7H2 both in solution and conjugated to AuNPs encouraged their evaluation in vivo. It was observed that minimal doses of peptides could be used when conjugated to AuNPs in order to observe a significant antimetastatic effect in comparison with the use of free peptides in solution. The antitumor effects were observed in the syngeneic metastatic model of B16F10-Nex2 murine melanoma in C57Bl/6 mice. At the end of the treatment, the animals had their lungs extracted and the tumor area was measured by analyzing the images of the lungs. It was observed that minimal doses of peptides could be used when conjugated to AuNPs in order to observe a significant antimetastatic effect in comparison with the use of free peptides in solution. It is noteworthy that the dose of each peptide was 50 μg both free in solution and conjugated to AuNPs. The 50 μg dose of each peptide is a value 5× lower than the dose at which these peptides presented antitumor results in vivo in the work previously published by Polonelli et al. [23,24]. Therefore, by using a dose much lower than that already used, we proved the capacity of AuNPs to increase the therapeutic efficiency of peptides.

Despite the considerable progress achieved in the development of chemotherapeutic agents based on AuNPs, most of these systems focus their results on in vitro profiles. The in vivo administration of AuNPs functionalized with therapeutic peptides is still very restricted or addresses the use of targeting peptides conjugated to NPs together with a drug, aiming at the active targeting mechanism in which the peptides interact with specific membrane receptors present in tumor cells [14,44,45]. For example, Chanda et al. conjugated analogs of the bombesin peptide (BN) to AuNPs [44]. BN is a peptide that can specifically interact with local receptors on cancer cells, called gastrin-releasing peptides (GRP). It is known that BN peptides have a high affinity for GRP receptors that are overexpressed in breast, prostate, and lung carcinomas. AuNPs-BN were administered intraperitoneally in mice bearing prostate tumors. It was reported that AuNPs-BN showed enhanced tumor accumulation compared to radiation-labeled free BN or unconjugated AuNPs. Thus, AuNPs functionalized with BN, improved the target specificity of prostate tumor cells. In another reported study, spherical AuNPs conjugated with lcosine-I, a peptide isolated from the spider’s venom, endowed with cell penetration and cytotoxicity against cancer cells, had an efficiency of absorption of AuNPs approximately 8 times higher than that of AuNPs pure, exhibiting efficiency of cell internalization, selectivity, and toxicity against cancer cells [45].

## 5. Conclusions

The functionalization of AuNPs with C7H2 and HuaL1 peptides was shown to be a clever strategy to promote their simultaneous therapeutic activity, resulting in the inhibition of melanoma metastasis in the animal model. Although the antitumoral and antimicrobial activities of both peptides had been reported before individually, herein they were shown to be efficient at lower concentrations when simultaneously conjugated to AuNPs. In the development of antitumor drugs, the efficiency of the drugs at low doses is very important as it reduces the occurrence of side effects. We believe that our results were very encouraging and favorable to the application of peptides-conjugated AuNPs in cancer therapy and that this strategy could be extended to the use of other bioactive peptides.

## Figures and Tables

**Figure 1 pharmaceutics-14-01324-f001:**
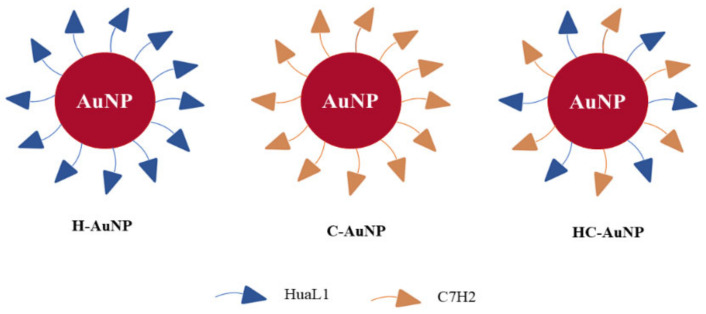
Schematic representation of peptides-conjugated AuNPs obtained after the NPs’ functionalization with the peptides C7H2 and HuaL1 in isolation or simultaneously.

**Figure 2 pharmaceutics-14-01324-f002:**
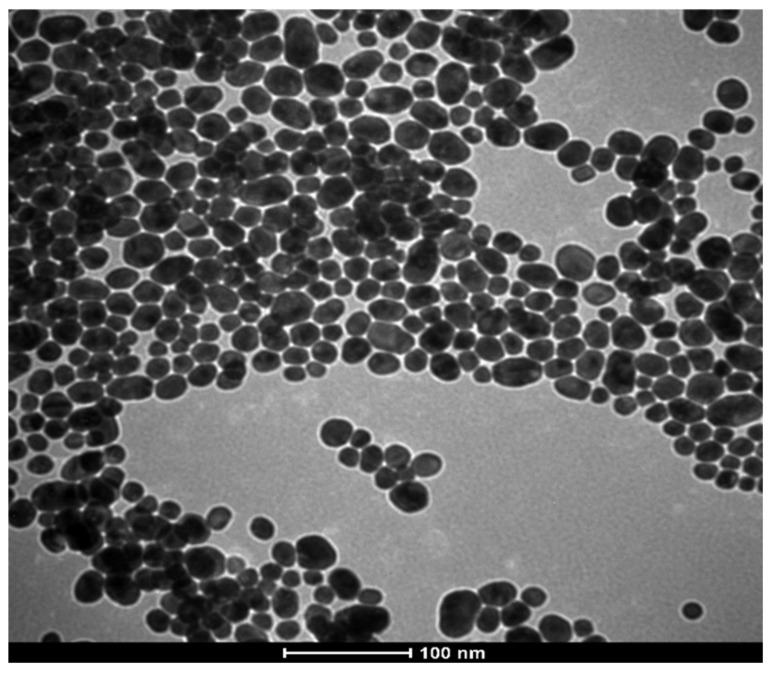
TEM image of AuNPs.

**Figure 3 pharmaceutics-14-01324-f003:**
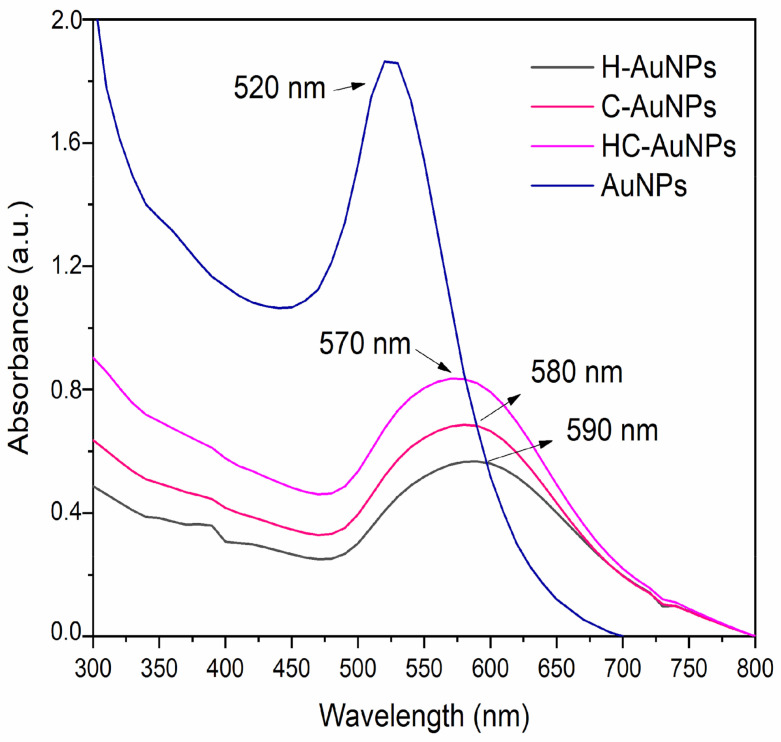
UV-vis spectra of AuNPs before and after the conjugation with HuAL1 and C7H2 peptides.

**Figure 4 pharmaceutics-14-01324-f004:**
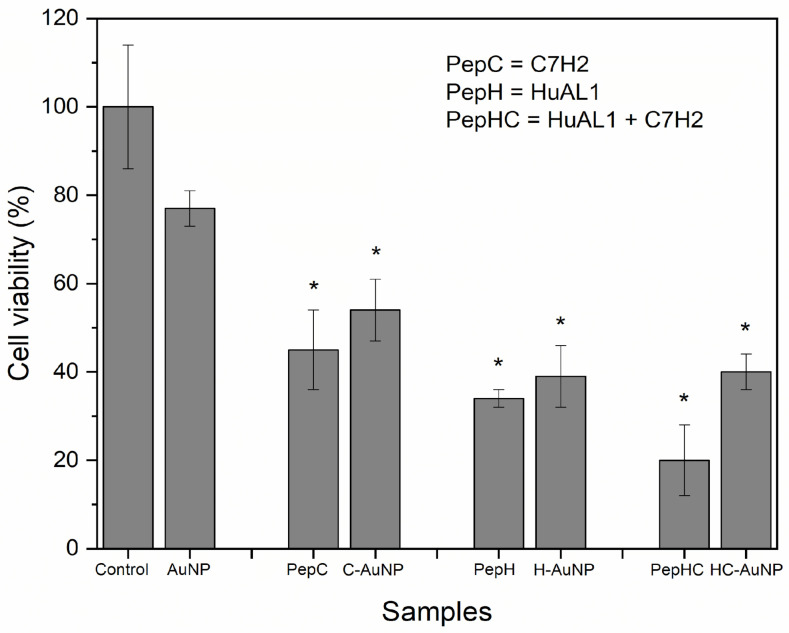
Viability of Hs68 cells incubated with AuNPs before and after conjugation with both peptides C7H2 and HuAL1, individually (C-AuNP and H-AuNP) or combined (HC-AuNP). The cells were also incubated with the peptides free in solution. Values are depicted as mean values and standard deviation (*n* = 3) (* *p*-value < 0.001).

**Figure 5 pharmaceutics-14-01324-f005:**
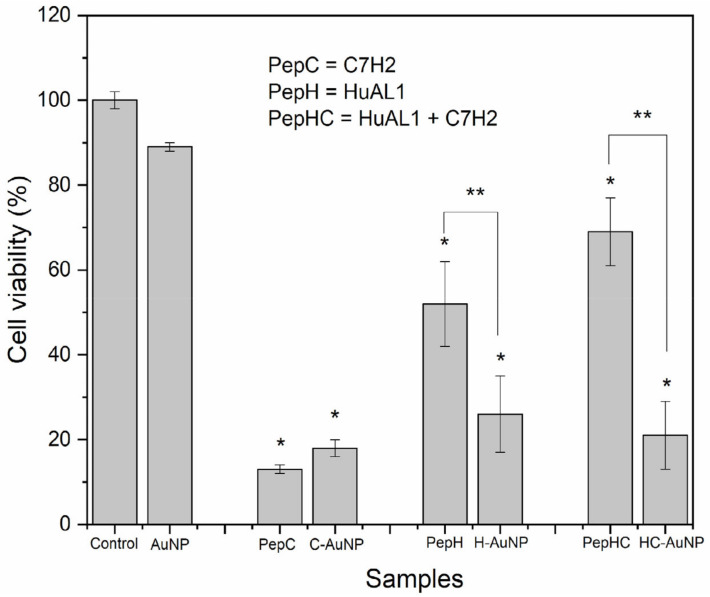
Viability of B16 cells incubated with AuNPs before and after conjugation with both peptides C7H2 and HuAL1, individually (C-AuNP and H-AuNP) or combined (HC-AuNP). The cells were also incubated with the peptides free in solution. Values are depicted as mean values and standard deviation (*n* = 3) (* *p*-value < 0.001 in comparison with the control group; ** *p*-value < 0.001 in comparison with each other).

**Figure 6 pharmaceutics-14-01324-f006:**
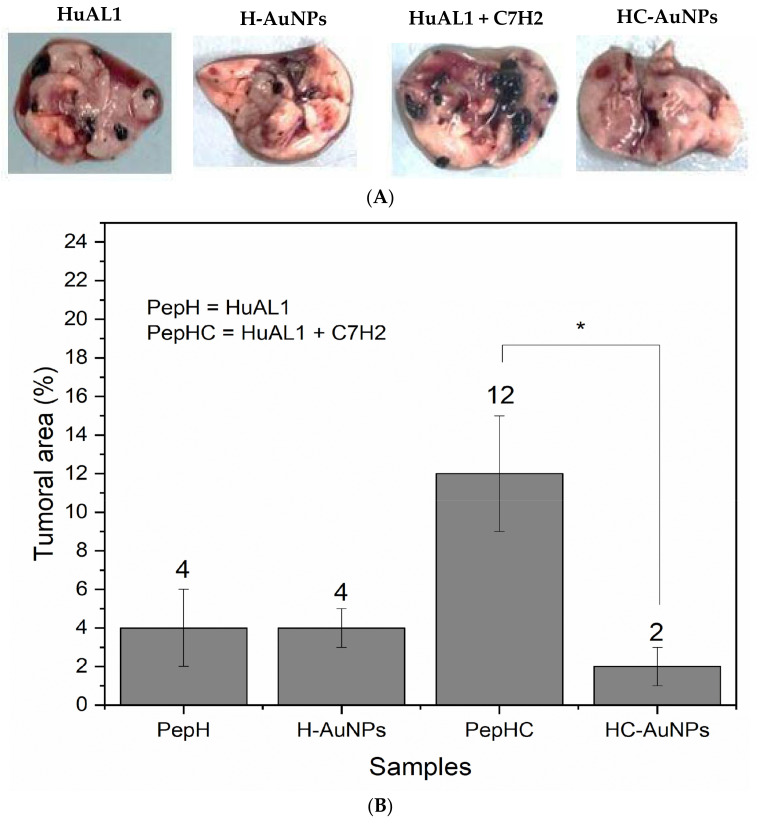
Image (**A**) and representation of the tumoral area (**B**) of the lungs extracted from the animals treated with HuAL1 individually or combined with C7H2, H-AuNPs, and HC-AuNPs (* *p* < 0.001).

**Figure 7 pharmaceutics-14-01324-f007:**
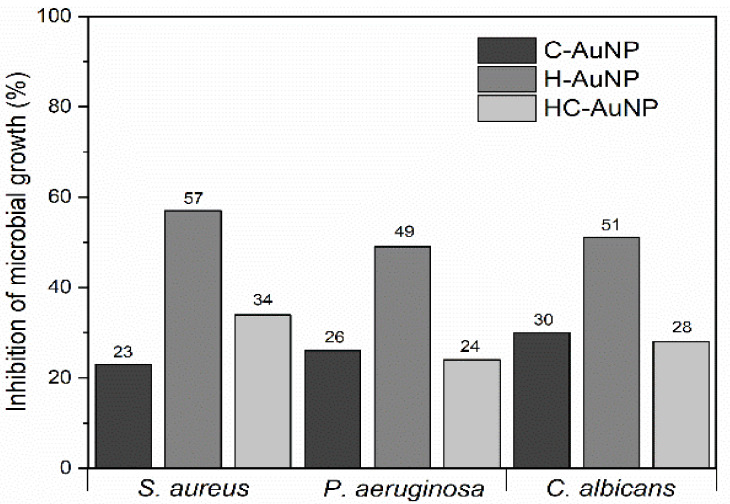
Inhibition of microbial growth after incubation with peptides-conjugated NPs.

**Table 1 pharmaceutics-14-01324-t001:** Final concentration of peptides in the conjugated AuNPs suspension.

Samples	Peptides	Concentration of Peptides
H-AuNPs	HuAL1	1 mM
C-AuNPs	C7H2	1 mM
HC-AuNPs	HuAL1 + C7H2	1 mM

**Table 2 pharmaceutics-14-01324-t002:** Concentration of peptides per-well for each sample.

Samples	Concentration (mM)	
	HuAL1	C7H2
H-AuNPs	0.6	0
C-AuNPs	0	0.6
HC-AuNPs	0.3	0.3

**Table 3 pharmaceutics-14-01324-t003:** Concentration of peptides after several dilutions of peptides-conjugated NPs.

Volume of NPs Per-Well (µL)	Concentration of Peptides (mg/mL)
120	1.20
100	1.00
60	0.60
30	0.30
15	0.15

**Table 4 pharmaceutics-14-01324-t004:** Hydrodynamic diameter, polydispersity index, and zeta-potential of AuNPs before and after conjugation with peptides.

Samples	Hydrodynamic Diameter (nm)	Polydispersity Index (PDI)	Zeta-Potential (mV)
AuNPs	23 ± 8	0.22	−36 ± 5
H-AuNPs	281 ± 7	0.28	3 ± 1
C-AuNPs	239 ± 15	0.30	3 ± 2
HC-AuNPs	270 ± 22	0.32	3 ± 1

**Table 5 pharmaceutics-14-01324-t005:** Concentration of peptides that was able to cause the inhibition of microbial growth.

Microorganism	Concentration of Peptides (mg/mL)		
	C-AuNPs	H-AuNPs	HC-AuNPs
*S. aureus*	0.6	1.0	0.6
*P. aeruginosa*	0.6	1.2	1.2
*C. albicans*	0.6	1.2	1.2

**Table 6 pharmaceutics-14-01324-t006:** Metal concentration related to the maximum antimicrobial activity of peptides-conjugated NPs.

Microorganism	Concentration of Metal (µg/mL)		
	C-AuNPs	H-AuNPs	HC-AuNPs
*S. aureus*	113.4	189.0	113.4
*P. aeruginosa*	113.4	226.8	226.8
*C. albicans*	226.8	226.8	226.8

## Data Availability

Raw data were generated at ICT-UNIFESP and UMC. Derived data supporting the findings of this study are available from the corresponding authors D.B.T and D.C.A. on request.

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
