# Peer review of "Co-Functionalization of Gold Nanoparticles with C7H2 and HuAL1 Peptides: Enhanced Antimicrobial and Antitumoral Activities"

_pharmaceutics, 2022, doi:10.3390/pharmaceutics14071324_

Round 1

Reviewer 1 Report

Review on manuscript pharmaceutics-1757783

The article is about the preparation of peptide-functionalized gold NPs and their biomedical application. The topic is good, but the authors should revise some part before publication. 

- the purity of the applied chemicals is missing 

- the desciprtion of the equipments is poor. e.g. DLS studies: what is the wavelength of the laser? Why the Smolusovki equation was used for Zeta potencial calculation? Is it valid for small NPs in the absence of ionic strength?   

- How did the authors confirm the surface coverage of the NPs?? The surface coverage has important effect....

- the Zeta-potencial values have also standard deviation

-I recommend revision 

Reviewer 2 Report

The manuscript about antimicrobial and antitumoral action of some functionalized gold nanoparticles has as subject a known area of research that is still attracting scientists. The functionalization was performed with two peptides, and the hybrid nanoparticles were characterized by TEM, DLS and ICP. The work is well described, with some flaws that are detailed below. 42 references ends up the manuscript; insertion of DOI numbers will be an asset, and also their arrangement without empty lines between them. The Abstract seems a little bit too long.

-line 83- no citation of Arruda work;

-are the peptides used degraded by simple gold nanoparticles? how can be evaluated their activity after functionalization with gold nanoparticles- a direct comparizon should be provided;

-no citation on line 105;

-line 126 and 128- some chemical words do not need letters in capital;

-the role of EDC and NHS used in the functionalization is not described;

-that is the charging of peptides on gold nanoparticles? what is the proof that peptides are indeed linked to gold nanoparticles? while the concentration of gold was measured by ICP as 1.9 mM, what is the concentration of peptides?

-Table 1,2,3 and Figure 1 should be centered on page, and so on;

-TEM picture is of gold nanoparticles functionalized or not? two pictures are necessary for comparizon, as UV-VIs is not a test for functionalization, the change could be just becoase of agglomeration.

The conclusion chapter should be re-writen.

Finnaly, the work needs a real and careful improvement before acceptance for publication in a high-ranked journal.

Round 2

Reviewer 1 Report

Accept

Reviewer 2 Report

The authors took into consideration all the requested issues and improved their manuscript to a high extent, which makes it suitable for publication. Lines 722-724 should be removed and the main text formatted as the template of the journal shows.